# Ecotoxicity of Heteroaggregates of Polystyrene Nanospheres in Chironomidae and Amphibian

**DOI:** 10.3390/nano12152730

**Published:** 2022-08-08

**Authors:** Florence Mouchet, Laura Rowenczyk, Antoine Minet, Fanny Clergeaud, Jérôme Silvestre, Eric Pinelli, Jessica Ferriol, Joséphine Leflaive, Loïc Ten-Hage, Julien Gigault, Alexandra Ter Halle, Laury Gauthier

**Affiliations:** 1Laboratoire d’Ecologie Fonctionnelle et Environnement, UMR 5245 CNRS, Université Paul Sabatier, 31062 Toulouse, France; 2Laboratoire des Interactions Moléculaires et Réactivité Chimique et Photochimique (IMRCP), UMR CNRS 5623, Université Paul Sabatier, Bâtiment 2R1, 3ème étage, 118, Route de Narbonne, 31062 Toulouse, France; 3Laboratoire Takuvik, CNRS, Laval University, Avenue de la Médecine, Quebec, QC 1045, Canada

**Keywords:** nanoplastics, freshwater, heteroaggregation, (geno) toxicity, amphibians, dipters, humic acid

## Abstract

Due to their various properties as polymeric materials, plastics have been produced, used and ultimately discharged into the environment. Although some studies have shown their negative impacts on the marine environment, the effects of plastics on freshwater organisms are still poorly studied, while they could be widely in contact with this pollution. The current work aimed to better elucidate the impact and the toxicity mechanisms of two kinds of commercial functionalized nanoplastics, i.e., carboxylated polystyrene microspheres of, respectively, 350 and 50 nm (PS350 and PS50), and heteroaggregated PS50 with humic acid with an apparent size of 350 nm (PSHA), all used at environmental concentrations (0.1 to 100 µg L^−1^). For this purpose, two relevant biological and aquatic models—amphibian larvae, *Xenopus laevis*, and dipters, *Chironomus riparius*—were used under normalized exposure conditions. The acute, chronic, and genetic toxicity parameters were examined and discussed with regard to the fundamental characterization in media exposures and, especially, the aggregation state of the nanoplastics. The size of PS350 and PSHA remained similar in the *Xenopus* and Chironomus exposure media. Inversely, PS50 aggregated in both exposition media and finally appeared to be micrometric during the exposition tests. Interestingly, this work highlighted that PS350 has no significant effect on the tested species, while PS50 is the most prone to alter the growth of *Xenopus* but not of Chironomus. Finally, PSHA induced a significant genotoxicity in *Xenopus*.

## 1. Introduction

Plastic is a synthetic polymeric material that has been produced by humans since the early 20th century from hydrocarbons of the petrochemical industry. It is physically (heated, extruded, molded, etc.) and chemically (mixed with additives) worked to obtain a complex formula that serves its final purpose. Among the many different kinds of manufactured plastic, polypropylene (PP), high- and low-density polyethylene (HDPE and LDPE), polyethylene terephthalate (PET), and polystyrene (PS) [1] are the most produced in terms of tonnage.

Due to their numerous properties as raw materials or polymers, plastics have been massively produced and used for decades in everyday life and industrials activities [2,3]. Recent estimations indicated that the global production of plastic exceeds 250–200 million tons per year [4], passing from 1.5 million tons in 1951 to around 335 million tons in 2016 [5] to reach a total of 359 million tons in 2018 [6]. This is without counting the current increasing production, use and consumption of single-use plastic items due to the COVID-19 pandemic [7,8], which may have greatly expanded these data.

Considering the life cycle of each of these items, it is not surprising to find out that plastic is distributed all over the world, especially in the aquatic compartment, which concentrates all forms of pollution. Lately, the efforts of researchers have focused on the potential effects of these plastics in the marine aquatic environment in particular. Numerous studies have shown the negative impacts of plastic debris ingestion on various marine organisms, leading to deleterious effects, such as on reproduction, fertilization, immunity, and oxidative stress [9,10,11,12].

What about freshwater systems? The question naturally, indeed, arises for freshwater aquatic systems, which are interconnected. Yet, the subject is much less informed. The paper of Emmerik and Schwartz [13] describes the state of the art concerning the presence of plastics in rivers and identifies several sources and transportation pathways. Moreover, recent models of plastic debris transportation based on the global circulation of currents have estimated that rivers are the main land-to-sea pathway [14,15]. There is a considerable gap regarding the concentration of nanoplastics in environmental samples due to a lack of specificity and/or resolution of the current analysis methodologies. Only Sullivan et al. [16] published semi-quantitative data about polystyrene nanoplastics in the river Tawe (South Wales, 241.8 μg·L^−1^). Impacts on freshwater organisms are still poorly studied while they appear at the forefront of this environmental disaster. Indeed, although some studies have proven the consumption of microplastics by riverine organisms, such as bivalves [17], their potential effects are yet to be shown.

Concerning the assessment of the nanoplastics’ ecotoxicity, the question is not simple. Their toxicity to living organisms may depend on many factors, including particle size. Size is the established factor that allows for their classification. It is a function of the fragmentation and erosion in the environment [18]. The main weathering processes involved are mechanical stress and oxidation upon solar light [19]. Macroplastics of 5 cm and more become smaller and smaller: debris of 5 mm–5 cm are mesoplastics, those of 0.1 µm–5 mm are microplastics, and nanoplastics are under 0.1 µm, the only one invisible to the naked eyes [20]. While large fragments can pose apparent problems in organisms, sometimes causing mortality by choking, ingestion and entanglements, the effects of smaller pieces may be more insidious, and their toxicity may go unnoticed. The formation of micro and nanoplastic is of great concern since they can be available for many organisms within the marine environment [21] and the aquatic food web [22]. From an ecotoxicological point of view, as with all nanoparticles, their ecotoxicity becomes more complex as plastic materials degrade. The ecotoxicity of nanoplastics is more questionable since, as colloidal materials, they present a high surface-area-to-volume ratio that enhances their reactivity [23] and since their nano size is theoretically compatible with their crossing of biological barriers as described by [24]. Another essential toxicity factor is the potential functionalization of the plastic debris, which is the case for all nanoparticles [25]. Therefore, it is now known that surface functionalization has a positive influence on the nanosphere’s toxicity [26,27,28]. Once in the environment during their transportation, nanoplastics may interact with Dissolved Organic Matter (DOM) (because of their larger surface–volume ratio) and further complicate their life story [29]. Interaction can affect their mobility, stability, and distribution in a water column [30]. Components of Organic Matter (OM) may be rapidly absorbed by particles in layers, forming their heteroaggregation and controlling their final fate, reactivity, and finally impact organisms [31,32].

In this context, the current work aimed to better elucidate the aquatic impact and the toxicity mechanisms of two kinds of commercial functionalized nanoplastics, i.e., carboxylated polystyrene microspheres of, respectively, 350 and 50 nm (PS350 and PS50), and 350 nm heteroagragates (PSHA) made of PS50 and a known amount of humic acid from the American Suwannee River (which constitutes a reference of riverine systems in ecotoxicological studies of nanoparticles using OM). The carboxylate surfaces were chosen in order to address the natural weathering of the nanoplastics and the studied concentrations are of the order of µg L^−1^ and aim to be similar to those found in natural environments.

A previous study evaluated the toxicity of the exact same particles towards freshwater algae (four benthic and one planktonic algal species) in regard to particle states in the media [32]. Despite the very low toxicity exhibited by the algal species, the functioning of aquatic ecosystems may be destabilized because of the competitive interactions between species. For example, the strong growth of the planktonic algae (more than 150% with the heteroaggregates at 1 µg L^−1^) may reduce the growth of benthic algae through the reduction in light transmission in water. The conclusion was that the direct effects of plastic particles on primary consumers could also modify the top-down control of planktonic and benthic microalgae. Here, the question is to evaluate the toxicity of these identical plastic particles on primary consumers, but at another trophic level, one living in the water column and the other at the water–sediment interface. For this purpose, two relevant aquatic biological models—the dipter *Chironomus riparius* larvae and the amphibian *Xenopus laevis* larvae—were used under normalized exposure conditions [33,34]. Because of its position in food webs and its potential capacity to accumulate contaminants, Chironomus is an organism of choice for ecotoxicological studies. *Xenopus* larvae are also a good model for demonstrating acute, chronic, and genetic toxicity effects. Their exclusively aquatic lifestyle and the permeability of their skin lead to direct and constant exposure to contaminants present in aquatic environments to which they are sensitive. They have large and numerous chromosomes that are easily accessible to erythrocytes [35]. Key toxicities parameters (acute, chronic, and genetic endpoints) were examined and discussed concerning the fundamental characterization in media exposures of the formation of heteroaggregates of plastics. Aggregation mechanisms are imperative to elucidate to accurately and representatively evaluate their toxic effect in the laboratory [23].

## 2. Materials and Methods

### 2.1. Plastic Materials and Physical-Chemical Analyze

#### 2.1.1. Raw Materials, Preparation of Heteroaggregates, and Physical Characterizations in Water

Materials, heteroaggregates preparation, and fundamental characterizations/methods were described in detail in Rowenczyk et al. [32]. Table 1 summarizes the characteristics of the particles in water. Briefly, a dispersion of PS50 in water (250 mg L^−1^, carboxylate polystyrene microspheres, Polybead^®^, Polysciences, Warrington, PA, USA) and a solution of Humic Acids (HAs) in water (0.25 g L^−1^, Cat#2S101H, Standard II, International Humic Substances Society, USA, pH 7.20 ± 0.05 using HCl) were mixed (1:1 *v*/*v*) to obtain the PSHA suspension (100 mg L^−1^) as follows. The solution’s ionic strength was increased to favor the heteroaggregation by adding 50 mL of NaCl (3.5 mol. L^−1^) to reach the concentration of 700 mmol. L^−1^ (ionic strength, I = 700 mmol. L^−1^). The concentration of the bulk PS-HA suspension (100 µg L^−1^) was diluted 1000 times prior to exposure tests; then, the salt’s final residual concentrations were 0.7 mmol. L^−1^, which is significantly low compared to the ionic strength of the biological media (Appendix A). Thus, we assumed that this slight salt variation would not affect the biological essays. From DLS measurements in water, it has been concluded that, on average, the primary sizes of PS350 and PSHA were similar, while the those of PS50 was around 10 times smaller. All the characteristics of the raw particles are summarized in Table 1.

#### 2.1.2. Physical Characterization in Exposure Media

Characterizations of the particles were also carried out in the two media of exposure (those of *Chironomus riparius* and *Xenopus laevis*) thanks to DLS measurements and TEM observations (Hitachi HT7700, Tokyo, Japan). For the TEM observation, a droplet of 20 µL of the sample was adsorbed on a discharged collodion/carbon-coated copper grid. After one minute, the grid was stained for 10 s by inverting onto a drop of 2% uranyl aqueous solution. The grid was blotted using filter paper before imaging TEM operating at 80 kV.

*Xenopus* media was recovered after exposure to the organisms and filtrated on PES 0.2 µm. This media was characterized (pH and I), then particles were added at a concentration of 10 mg L^−1^. These dispersions were characterized by DLS (Cumulant method) and zetametry.

Since PS-HA could potentially disaggregate during exposure, the stability of these heteroaggregates was tested by diluting the PSHA solution at 700 mM NaCl into water (decreasing the ionic strength) to ensure their dilution in the media could not change their morphology. Microsphere surface charges were measured through the zeta potential (ZP) determination in the media at pH 7. Functionalization by carboxylic moieties was evaluated by micro-FTIR.

### 2.2. Organisms, Breeding, and Toxicity Assessment

#### 2.2.1. Chironomids

*C. riparius* larvae were obtained from the breeding that is maintained at the laboratory of functional ecology and environment under conditions complying with standards [34,36,37,38]. Ten larvae of 48 h old per condition and seven replicates were exposed to a range of concentrations of PS50, PS350, and heteroaggregates PS50-HA (0, 0.1, 1, 10, and 100 µg L^−1^). Concentrations of plastics were obtained from stock solutions prepared in sterile ultrapure water at 1 mg and 100 mg L^−1^. The water column brought the contamination in each condition at the beginning of the exposure. Larvae were exposed in the static condition in 300 mL glass beakers (Pyrex^®^) containing sediment and reconstituted water RW_chiro_ (66.2 mg L^−1^ CaCl_2_.2H_2_O; 61.4 mg L^−1^ MgSO_4_.7H_2_O; 96 mg L^−1^ NaHCO_3_; 4 mg L^−1^ KCl; 63 mg L^−1^ CaSO_4_, 2H_2_O; 1 mg L^−1^ NaBr) as described in the AFNOR standard T 90 339-1 [37]. The negative control consisted of reconstituted water. Additional control was performed with HA (100 µg L^−1^) in RW_chiro_ at a concentration equivalent to the one of the PSHA 100 µg L^−1^ condition to exclude the role of HA in toxicity assessment. Exposure was carried out at 21 ± 1 °C, with gentle aeration, under 16:8 light–dark cycles. The larvae were fed daily with fish food solution (Tetramin^®^, Toulouse, France). At the end of the exposure, mortality was recorded, and growth inhibition was determined by assessing survival and measuring the total body length of larvae (ImageJ^®^, software, v 1.8.0, editor Bharti Airtel Ltd., Department of Health and Human Services (HHS), Washington, DC, USA). The cephalic capsule of each larva was also measured to determine larval instars [36] and assess potential delays in development. Teratogenicity was evaluated on cephalic capsules as described by Dias et al. [39]. Mouthpart deformities were assessed and rated according to Warwick and Tisdale [40] and Vermeulen et al. (1998) [41]. Treatment groups were compared based on their rating (seriousness of deformities) and probability of occurrence of deformities (individual or total deformities). Mortality, organism sizes, and teratogenicity were compared with a Kruskal–Wallis test, followed by Dunn’s test to analyze differences between groups. The probability of mouthpart deformities in chironomids was determined with the chi-square test. All analyses were performed using Sigma Plot 12.0 software (Inpixon Systat Software, Palo Alto, CA, USA).

#### 2.2.2. Xenopus

*X. laevis* larvae were obtained and grown at the laboratory as described in Mouchet et al. [42,43]. Groups of 15 *Xenopus* (Stage 50, [44]) were exposed for 12 days in semi-static conditions to the same range of concentrations of the same plastic particles, PS50, PS350, and heteroaggregates PS50-HA (0, 0.1, 1, 10 and 100 µg L^−1^), used for the experimentations in Chironomus. Concentrations of plastics were also obtained from stock solutions prepared in sterile ultrapure water at 1 and 100 mg L^−1^. Larvae were exposed to daily exposure media renewal and feeding (Tetraphyll^®^) according to standardized procedures [33]. The negative control (NC) condition was composed of reconstituted water RW_xeno_ only (294 mg L^−1^ CaCl_2_ 2H_2_O; 123.25 mg L^−1^ MgSO_4_ 7H_2_O; 64.75 mg L^−1^ NaHCO_3_; 5.75 mg L^−1^ KCl) and positive control (PC) was composed of RW_xeno_ added with cyclophosphamide monohydrate ([6055–19–2], Sigma-Aldrich Chimie, Saint-Quentin Fallavier, France) at 40 mg L^−1^. The same HA control was carried out to 100 µg L^−1^, corresponding to the maximal concentration present in PSHA heteroaggregates.

At the end of exposure, mortality, growth inhibition, and genotoxicity were assessed as follows: (i) mortality was examined by counting dead animals and expressed as a percentage, and (ii) chronic toxicity (growth inhibition) was evaluated by measuring the total body length of each larva at the beginning of the exposure and the end of the exposure (analysis software [45]); statistical analyses were performed on organisms’ size using a Kruskal–Wallis test, followed by Dunn’s test to analyze differences between groups. For the sake of clarity, the graphic representation is based on Growth Rates (GRs), calculated for each group as follows: $$\mathrm{GR}=\frac{\mathrm{AG}-\mathrm{negative}\;\mathrm{control}\;\mathrm{AG}}{\mathrm{negative}\;\mathrm{control}\;\mathrm{AG}}\times 100$$
with AG: average growth within a treatment group, determined as the difference between the average size at the end and the average size at the beginning of the experiment. (iii) Genotoxicity was performed on blood samples obtained from anesthetized larvae (tricaine methane sulfonate, Sigma-Aldrich Chimie S.a.r.l, Saint-Quentin-Fallavier France); the number of erythrocytes containing one micronucleus or more (micronucleated erythrocytes, MNE) was determined in a sample of 1000 erythrocytes per larva, and results were expressed as the number of MNE per thousand (MNE‰). The statistical method used is based on comparisons of medians by determining the theoretical median for each group [46]. The difference between the theoretical medians of the test groups and the theoretical median of the negative control group is significant if there is no overlap (95% certainty). In this case, micronucleus induction in exposed larvae is considered as a significant genotoxic response.

## 3. Results

### 3.1. Morphological Characterization of the Particles and Aggregates

The sizes of the dispersions of PS in the media were evaluated in the exposure media of both biological models. PS350 and PSHA did not aggregate in both exposure media and their size remained around 300–500 nm (Table 2). On the contrary, PS50 tended to homoaggregate in both exposure media and was observed thanks to DLS (Appendix A) measurements and TEM observations (Figure 1). In Figure 1, a fractal-like structure of the aggregated PS50 can be observed. Even if these micrometric structures cannot be correctly characterized in their three dimensions, these observations confirm the aggregation pathways. Moreover, in the media, PS50 reached a final size larger than the PS350 and the PSHA ones.

Since PSHA are aggregated materials, there is a probability that their dilution in the media could change their final morphology. The stability of these heteroaggregates was thus tested by diluting the PSHA at 700 mM NaCl in water (decreasing the ionic strength) (Appendix A). The PSHA size remained remarkably stable, around 300 nm, in the whole range of salt concentration tested. These results proved that by diluting the PSHA in the media, the size of the PSHA will not be modified (Appendix A).

Microsphere surface charges were measured through the zeta potential (ZP) determination in the media at pH 7 (Table 3). In both media, we observed that the charges were significantly higher for PS350 than for PS50. This can be explained because the PS350 is more functionalized by carboxylic moieties, as measured by micro-FTIR. For example, the PS350 zeta potential in the amphibian medium was equal to −30 (±1.1) mV, significantly more intense compared to the charges of −16.5 (±0.3) mV measured for PS50. Thereby, the electrostatic repulsions could explain why PS350 presents higher stability towards homoaggregation compared to PS50. The charges of PSHA in the amphibian medium, equal to −23.0 (±0.5) mV, were between the two previous tested microspheres. The zeta potential differences from one media to another were insignificant (Table 3).

### 3.2. Toxicity Assessment on Biological Models

#### 3.2.1. Toxicity on Chironomids

After 7 days of exposure in static exposure, no mortality was observed up to 100 µg L^−1^ (data not shown). No growth inhibition was observed on larvae exposed to PS50 and PS350 regardless of the concentration (Figure 2), to the lowest PSHA concentrations and to the HA control. In contrast, larvae exposed to PSHA from 10 µg L^−1^ showed a significant growth inhibition (>10%) compared to the negative control. Similarly, cephalic capsule measurements also revealed a significant delay in the development of larvae exposed to PSHA from 10 µg L^−1^ (<26% of larvae at stage 4 compared to the negative control to 10 µg L^−1^ and <11% to 100 µg L^−1^) (Figure 3). The study of mouthpart deformities revealed no teratogenicity on Chironomus. Group comparison based on ratings revealed no teratogenicity whatever the concentration or types of aggregates. Groups were also compared based on the probability of deformity occurrence, and no difference between groups was highlighted with this method either (Table 4).

#### 3.2.2. Toxicity on Amphibians

The results show no mortality of *Xenopus* larvae exposed to PS50, PS350, or PSHA after 12 days of exposure (data not shown). Significant growth inhibitions are observed in larvae exposed to PS50, regardless of the concentration, from 10 µg L^−1^ of PS350 and to 100 µg L^−1^ of PSHA (Figure 4). Nevertheless, these inhibitions are weak (less than 10%). Significant genotoxicity was only observed in larvae exposed to PSHA from 10 µg L^−1^ (Figure 5).

## 4. Discussion

During the exposure, no mortality was observed no matter the nature or the concentration of plastic particles for either *Xenopus* or *Chironomus*. However, mortality was not the expected effect during this study dealing with nanoplastics such as PS50, PS350, and PSHA. Indeed, direct mortality of organisms is mainly observed in the case of ingestion of larger particles of plastics (debris), which could lead to gut impaction or perforation. Moreover, in this study, we were interested in the toxic effect of polystyrene particles with regard to their size, while other deleterious effects, such as co-ingestion of a toxic compound or a cocktail of compounds adsorbed on environmental plastics, are not studied here. In any case, an increase in mortality is generally not observed in organisms exposed to such low concentrations (in the order of µg L^−1^) but rather in the order of grams per liter, as was the case for *Tubifex tubifex*, which was overexposed to 2 mg L^−1^ of microplastics [47]. Nevertheless, this work has highlighted other significant effects on *Chironumus* and *Xenopus*.

In the case of *Chironomus*, neither PS50 nor PS350 affect their growth or teratogenicity. Only PSHA induced significant, but low, growth inhibition at 10 and 100 µg L^−1^. The effect of these particles could be discussed in regard to their size and aggregation state. Although PS50 showed the smallest size in pure water, it also forms the largest aggregates in the media exposure. It could be suggested that a direct effect of these aggregates would be inhibited because of their large size (>1 µm) preventing their ingestion. PS350 (401 ± 8 nm), which is smaller than PS50, but similar to PSHA (528 ± 22 nm) in size, induced fewer effects than PSHA (528 ± 22 nm), similar in size to PS350 (401 ± 64 nm). Size alone, even if well known as one of the main parameters that control the toxicity, cannot explain the observed differences in effects. The results suggest that other parameters would consider influencing their biological interactions. In the same way, the nature of the groups identified at the surface of the particle in the pure water (Table 1) with a COOH/NH_2_ group for PS-HA and a single COOH group for PS50 and PS350 could suggest that NH_2_ would contribute to the media exposure, above a specific concentration, to growth inhibition and to delay the stage of development in *Chironomus*, without affecting teratogenicity. As the AH controls also carry the NH_2_ functions, interactions and suspension processes in the water column would finally have more effect than the aggregated NP, whatever their size.

For *Xenopus*, the measured effects are different. PS50 induced significant growth inhibition regardless of the concentration. PS350 and PSHA also caused considerable growth inhibitions, but at higher concentrations (i.e., 10 and 100 µg L^−1^ for PS350, and 100 µg L^−1^ for PSHA). In this case, PS50, which induces the most significant effect on the growth of *Xenopus*, is made of the smallest particles but forms the larger aggregates in the culture media (>1 µm). Indeed, size is known as one of the main parameters modulating particle toxicity. These effects have already been demonstrated in *Xenopus* in the case of exposure to carbon nanoparticles, for example [48,49]. This toxicity can be qualified as direct when it is mechanical or indirect when it is nutritional. Most of the time, both direct and indirect types are probably expressed since they are linked, the first leading to the second. In the case of particles of plastic, Besseling et al. [50], for instance, observed a reduction in the body size of *D. magna* exposed to PS particles (70 nm in diameter) but from much higher concentrations (30–130 mg L^−1^). In the same way, Tang et al. [51] did not observe any mortality in *D. magna* exposed to PS particles (i.e., 1.25-μm; 2 to 8 mg L^−1^), but they also noted a reduction in body growth rate and increased transcription of arginine kinase and permease (enzymes involved in oxidative defense and energy production). In contrast, De Felice et al. [52] showed no alteration in growth in *X. laevis* larvae exposed to PS microparticles (3 µm in diameter) at higher concentrations (0.125 to 12.5 mg L^−1^). However, the test’s duration and the larvae’s development stage are different; the comparison is not suitable (stage 36 at the start of the 7-day test vs. stage 50 at the beginning of 12 days). Moreover, the PS beads used in De Felice et al. [52] have a much larger diameter than those used in the present study, 3 µm and 50–350 nm, respectively. However, in the present study, size alone cannot explain here the observed differences in terms of effects. These results suggest that other parameters could influence their biological impact, such as the structure, shape and surface of the aggregates/particles.

First of all, the chemical nature of the surface of the aggregates were compared. The chemical nature of the groups identified at the surface of the particle in the pure water were different (Table 1). While COOH/NH_2_ groups were found for PSHA surfaces, only COOH groups were identified for PS50 and PS350. These NH_2_ groups are likely originated from the AH. As AH controls did not exhibit any toxicity, it can be argued that the deleterious effect is due to the whole PSHA particle. This could suggest that these amides could be misrecognized as an extra source of energy by the organisms. Once ingested, PSHA could disaggregate in the insect gastrointestinal tractus and lead to important damages as observed by Matthews et al. on the flies *Drosophila melanogaster* [53]. This phenomenon could explain the growth rate reduction in the case of chironomes.

As a second approach, it is essential to comment on the behavior of the particles in each medium to understand their impact. PS50 is the particle with the lower surface charges in comparison to the two other particles (see ZP measurements). These charges are not sufficient to maintain the stability of the dispersion by electrostatic repulsion in the exposure media. This phenomenon results in a significant aggregation of PS50 during the exposure in both media, but at a different level. As PS50 is prone to aggregation in *Xenopus* exposure media, other nanomaterials undergo the same mechanism. Indeed, particle size is a critical factor affecting accumulation’s toxicity [54], and it is also the case for all particles that are non-soluble and behave like nano-objects.

Other mechanisms could also modify the size of the particles during the biological tests. While PSHA had a similar size in water and fresh media, it is enlarged when in aged exposure conditions. Conversely, PS50 is less aggregated in aged media than in the fresh media. Finally, the medium exposure of *Xenopus* did not impact the size of PS350 (Table 1). During the exposure, proteins, polysaccharides, carbohydrates, lipids, and so on, are produced by the organisms and are likely to adsorb the surface of the particles and form a corona. This phenomenon may have a fundamental role in the colloidal stability of particles. By changing their surface properties, size and density, these substances could destabilize the particle dispersion inducing aggregation and settling, for example, [55]. Moreover, the formation of a bio-corona could lead to misrecognition of the particles as energy sources, as already discussed.

In a general matter, the effects observed in terms of growth inhibition may be related to the particles of plastic ingestion in the intestinal tract. The literature reports that microplastic particles could be actually taken as prey for animals. This could induce their bioaccumulation, especially in the digestive tract as described by Ding et al. [56] and, therefore, a decrease in terms of food assimilation and body growth. Nanoplastics are expected to diffuse deeply into tissues and organs, crossing biological membranes. This could be the case for PS50 that induces significant inhibition of growth in *Xenopus*, even at low environmental concentrations (0.1 and 1 µg L^−1^).

Mechanisms of genotoxicity, which affect the integrity of the genetic material of cells, are different to those that induce growth inhibition. Genotoxicity may also be direct or indirect in organisms. The question of the direct nanogenotoxicity of nanoparticles comes down to whether the particle has crossed the membrane or not. In the case of indirect genotoxicity, this may be generated by immune, inflammation, or oxidative stress responses. For example, Li et al. [57] showed significant differences in differentially expressed genes between micro and nanoplastic exposure in *Corbicula fluminea* about the microbial disorder and the cascade pathway of the complement system induced by microplastics and with mucosal damages via the mitochondrial pathway in indirectly contacted tissues induced by nanoplastics. Some other authors demonstrated that micro- and nanoplastics induced no or low toxicity because they cannot cross cellular membranes in mammals [58].

Here, in this study, only PSHA is genotoxic at the highest concentrations from 10 µg L^−1^. It is true that PSHA is significantly smaller than PS50 in the culture medium of *Xenopus* (Table 2). Thus, this result would follow the principle of the toxicity of the small particles, which are known to be more bioavailable and, therefore, more toxic because of their large surface/volume ratio [59,60,61]. On the other hand, PSHA does not disaggregate during the processing of the toxicity tests and is finally the most aggregated in aged media of *Xenopus*, suggesting that the formed corona and COOH/NH_2_ groups could be implicated either directly or not in the genotoxic response to the highest concentrations. PSHA is also the most toxic polymer form against *Chironomus*. As already commented in this discussion, the presence of HA could increase the appetence of plastics leading to their grazing by *Xenopus* and *Chironomus.* Additionally, plastics are known to be palatable to microorganisms [62], and the plastisphere (mixed with the corona) would become even more palatable to these grazers.

The relatively low genotoxic response in *Xenopus* exposed to PSHA could be related to the fact that the OM that coats plastics may have a protective effect up to a certain limit. In the same way, Fadare et al. [63] showed that HA alleviates the toxicity of polystyrene nanoplastic particles to *Daphnia magna*. Some other authors found that HA has absorbed particles co-forming the corona without causing agglomeration or precipitation, finally leading to alleviated toxicity. In *Xenopus* larvae, the stomacal pH of around 2.5 [64] could modify the PS50 and HA interactions, allowing the bioavailability of more PS50 nanoparticles than when they are homoaggregated.

The measured effects are different from one species to another. Indeed, significant growth inhibition was observed in *Xenopus* exposed to PS50 whatever the concentration in contrast to *Chironomus* for which no significant inhibition is observed. In the same way, significant growth inhibition is also observed in *Xenopus* exposed to PS350 to the highest concentrations (from 10 µg L^−1^) and any growth inhibition in *Chironomus*. On the other hand, in *Chironomus*, PSHA induces a significant growth inhibition from 10 µg L^−1^ while it induced to only to the highest concentration of 100 µg L^−1^ in *Xenopus*. These results may suggest that the difference in sensitivity of both species may be concerning their habitat. Even if both primary consumers are grazers, *Xenopus* is more pelagic and spends most of its time in the water column. At the same time, the chironomid is associated with the sediment compartment at the interface water–sediment. PS50 homoaggregated in both media and would appear to be micrometric during the exposure. PS50 could be more available for *Xenopus* in the water column since PS50 would mainly be in the column water compared to the water–sediment interface. This result is surprising as the literature would suggest that for density reasons, PS microplastics may be available not only in the water column but also in the sediment, representing a higher risk for both pelagic and demersal organisms. In comparison, PE microplastics have a lower density, presenting a higher availability in the water column, and potentially pose a higher risk for pelagic organisms [65,66,67]. Furthermore, despite exposure to similar concentrations, the bioaccumulation kinetics parameters of nanoplastics (e.g., concentration factor, assimilation efficiency, elimination rates) from these two biological models are unknown and most likely different. The concentrations of nanoplastics found in their tissues would be distinct as well as detoxification processes, explaining the differences in effects and increased tolerance from one species to another.

This study allows one to compare the toxicity of three types of nanoplastics on three aquatic organisms under the same conditions. From a characterization perspective, the in situ DLS characterizations showed that it is not possible to predict the state of the particles in the media and that measuring their size in the media before, during, and at the end of the toxicity test is mandatory. To summarize the biological tests, the sensitivity of *Xenopus* was higher than that of *Chironomus*, which in turn was higher than that of algae [32], all exposed to the three same nanoplastics. Moreover, even if the positive effect on growth is not statistically significant in *Chironomus*, a tendency to increase is noted after exposure to 0.1 and 1 µg L^−1^ of PS50, as is the case for algae. The next step will be to evaluate the toxicity responses of organisms from different trophic levels (primary producers and consumers) within the same reconstituted environment in microcosms, allowing for measuring potential modulated toxicity.

## 5. Conclusions

This study evaluated the impact of polystyrene nanoplastics on two primary aquatic consumers, *Xenopus* and *Chironomus*. The low concentrations that have been carried out in this study make it original, as only a few studies were conducted at such environmentally relevant conditions. Nevertheless, these low levels of concentration may also explain why the recorded toxicity is mild in *Chironomus*. The results show that the effects of the same nanoplastics depend on the exposed organism and its characterizations in the media exposure (concentration, size, surface, aggregation state, etc.). The work ultimately underlines how difficult it is to accurately relate a biological response to a physicochemical characterization of nanoplastics. One of the further challenges to tackle would be to locate and quantify the nanoplastics in the exposed organisms and characterize them directly in a biological matrix. Overall, this study suggests that the toxicity evaluation of nanoplastics should be done conjointly with a solid physicochemical characterization of these particles.

## Figures and Tables

**Figure 1 nanomaterials-12-02730-f001:**
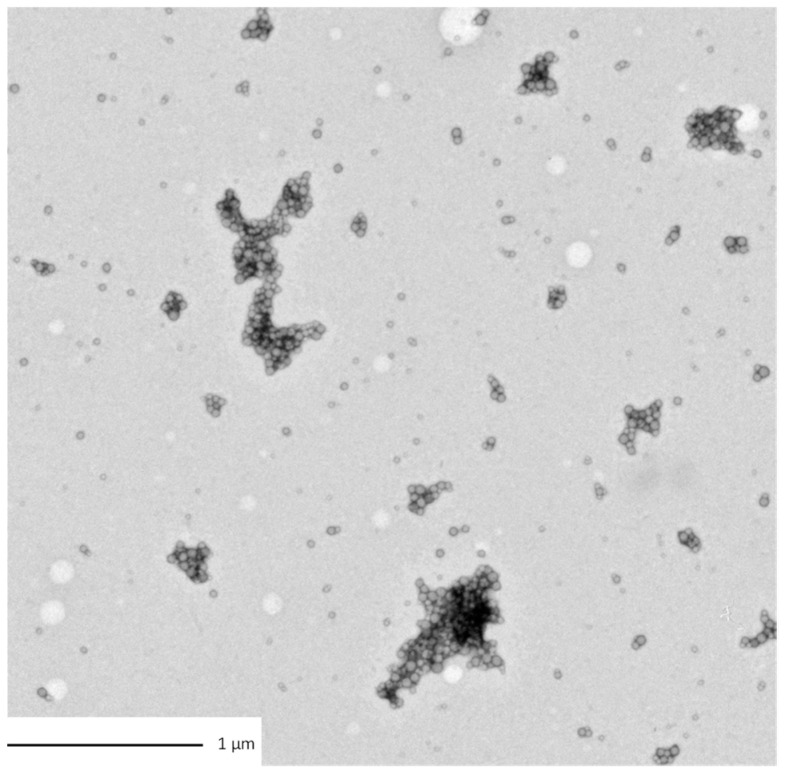
TEM observation of PS50 homoaggregation in Chironomus exposure media.

**Figure 2 nanomaterials-12-02730-f002:**
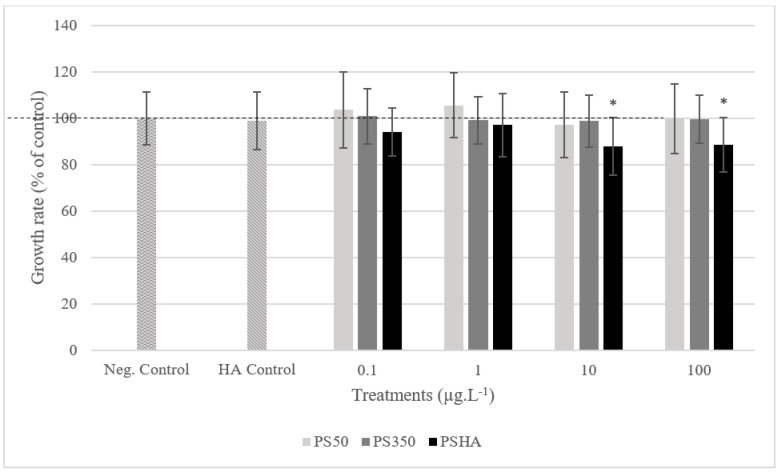
The growth rate of chironomid larvae exposed to PS50, PS350, and PS-HA. Error bars show the standard deviation of mean values. *: significantly different from control (Kruskal–Wallis test, *p* < 0.05).

**Figure 3 nanomaterials-12-02730-f003:**
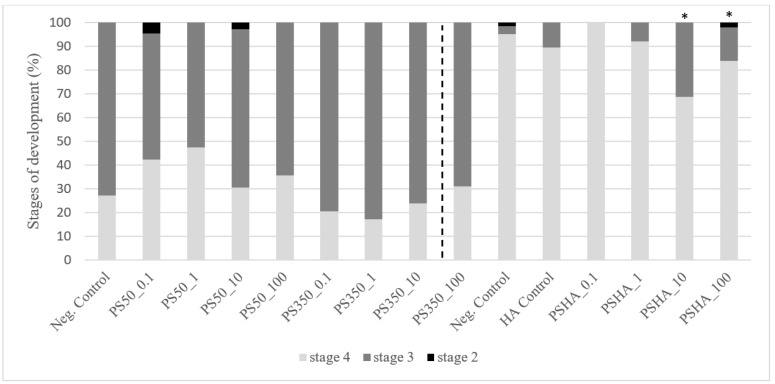
Stages of development of chironomid larvae exposed to PS50, PS350, and PS-HA. *: significantly different from their respective control (Kruskal–Wallis test, *p* < 0.05).

**Figure 4 nanomaterials-12-02730-f004:**
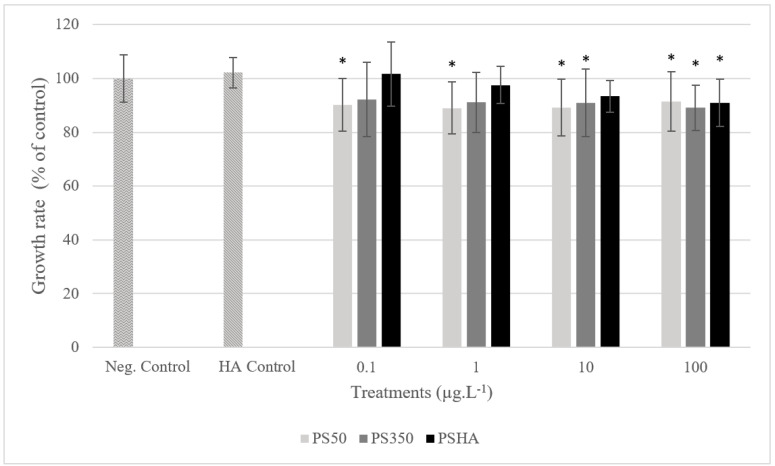
The growth rate of amphibian larvae exposed to PS50, PS350, and PSHA. Error bars show the standard deviation of mean values. *: significantly different from control (Kruskal–Wallis test, *p* < 0.05).

**Figure 5 nanomaterials-12-02730-f005:**
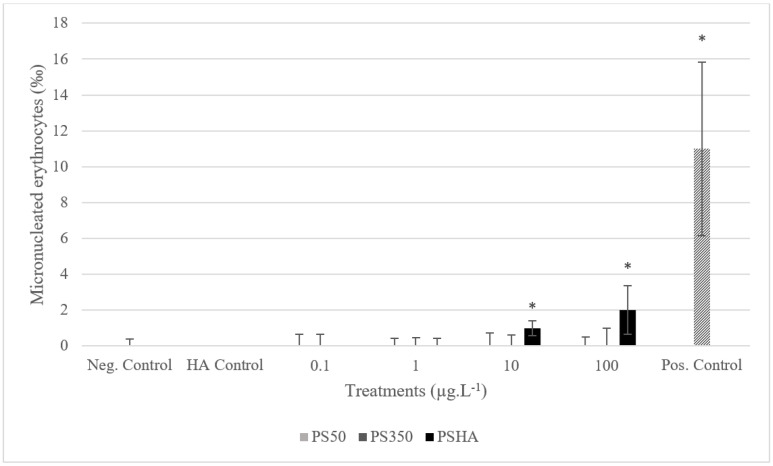
Genotoxicity of amphibian larvae exposed to PS50, PS350, and PSHA. Error bars show the 95 % confidence interval of median values. *: significantly different from control (based on comparisons of medians).

**Table 1 nanomaterials-12-02730-t001:** Physico-chemical characteristics of PS50, PS350, and heteroaggregates in pure water (for more details, see [32]).

Particles	PS50	PS350	PSHA
Primary size (nm)	50	350	300–500 in the medium in which they were designed (I = 700 mmol·L^−1^)
Transmission Electron Microscopy observation	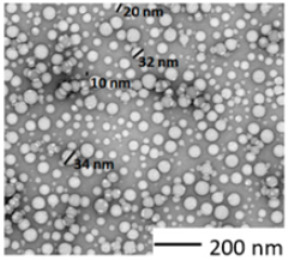	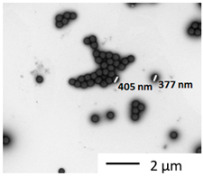	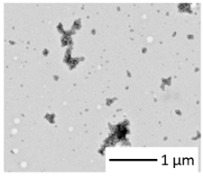
Polydispersity	Slightly polydisperse, made of particles from 10 to 35 nm	Monodisperse with a diameter between 370 and 410 nm	Polydisperse
Shape	Spherical	Spherical	Non-spherical
Nature of surface	–COOH	–COOH	–COOH/NH_2_
Surface charge	Negative	Negative	Negative
Functionalization	Less carboxylic moieties	More carboxylic moieties	

**Table 2 nanomaterials-12-02730-t002:** Size of the particles in the *Xenopus* and *Chironomus* exposure media (10 mg L^−1^).

Size Type	Medium	pH	PS50 dzH (nm)	PS350 dzH (nm)	PSHA dzH (nm)
Size in water	MilliQ	7.00	45 ± 1	349 ± 3	300–500 **
Size in media	*Chironomus*	8.23	2356 ± 294 *	401 ± 8	528 ± 22
Size in media	*Xenopus*	8.05	7255 ± 386 *	403 ± 10	484 ± 64
Size in media after exposure	*Xenopus* + (HA)	6.75	740 ± 124 *	414 ± 6.8	1053 ± 22 *

Since the PSHA size strongly depends on the preparation batch, we decided to put a range. * Out of the size range by DLS measurements. ** Measurement in water I = 700 mmol. L^−1^.

**Table 3 nanomaterials-12-02730-t003:** Potential zeta of particles in the exposure media after the pH adjustment at 7 (10 mg L^−1^). In brackets, the relative standard deviation values are reported.

Particles	Zeta Potential in Chironomus Exposition Medium (mV)	Zeta Potential in *Xenopus* Exposition Medium (mV)
PS50	−16.4 (±0.3)	−17.8 (±0.9)
PS350	−30.5 (±1.1)	−24.8 (±0.7)
PSHA	−23.0 (±0.5)	−25.5 (±2.8)

**Table 4 nanomaterials-12-02730-t004:** Teratogenesis on *C. riparius* larvae exposed to PS, PS350, and PSHA.

Polymer	Treatment (µg/L^−1^)	Deformity Frequency (%)
PS50	0	8.3
	0.1	11.1
	1	7.5
	10	11.1
	100	7.1
PS350	0	8.3
	0.1	4.5
	1	8.6
	10	10.9
	100	7.1
PSHA	0	0
	HA Control	3.4
	0.1	8.3
	1	0
	10	0
	100	6.9

## Data Availability

Not applicable.

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
