# Peer review of "Ecotoxicity of Heteroaggregates of Polystyrene Nanospheres in Chironomidae and Amphibian"

_nanomaterials, 2022, doi:10.3390/nano12152730_

Round 1

Reviewer 1 Report

This study, by Dr. Mouchet et al, is centered around the toxicity of polystyrene nanospheres on two aquatic models and demonstrate that PS50 most alters the growth of Xenopus but not Chironomus and that PSHA induced genotoxicity in Xenopus. However, as you mentioned in the discussion, it has been widely addressed that the difference in the physical properties of fine particles effect on the difference of biological response. In this regard, where is the novelty of this research? Moreover, data about the kinetics of each microplastics should be provided for considering whether it acts after being taken into the body. These data would lead to understand a risk for aquatic organism. Besides, I have some comments, which should be addressed.

1.      How much is the treatment concentration of present study compared to the concentration in the actual water environment? The rationale for setting the concentration should be stated. In addition, is there any report on the size of the microplastic that exists in the real environment?

2.      Since the difference in physical properties of nanoparticles affects the dispersibility, it is better to add the dispersive index in each exposure media. It is also known that the difference in shape affects the degree of toxicity. Thus, the author discusses the possibility of difference in growth inhibition due to the fact that PS-HA is non-spherical, and the others are spherical, if possible.

3.      It is desirable to be able to compare and discuss growth inhibition for Chironomids and amphibians more quantitatively by calculating the EC50 of each microplastic.

Author Response

This study, by Dr. Mouchet et al, is centered around the toxicity of polystyrene nanospheres on two aquatic models and demonstrate that PS50 most alters the growth of Xenopus but not Chironomus and that PSHA induced genotoxicity in Xenopus. However, as you mentioned in the discussion, it has been widely addressed that the difference in the physical properties of fine particles effect on the difference of biological response. In this regard, where is the novelty of this research?

This study takes into account the initial size, aggregation states of the particles before, during and after the exposure. While previous studies don’t always provide such information, it provides new insights allowing to understand the mechanisms.

Moreover, data about the kinetics of each microplastics should be provided for considering whether it acts after being taken into the body. These data would lead to understand a risk for aquatic organism.

We don't understand the recommandation. How to access these kinetic data please?

Besides, I have some comments, which should be addressed.

  1. How much is the treatment concentration of present study compared to the concentration in the actual water environment? The rationale for setting the concentration should be stated. In addition, is there any report on the size of the microplastic that exists in the real environment?

Below 1 to 10 µm, there are no published data. There are no quantitative data for nanoplastics in the environment, we based our concentration ranges on approximations/estimations.

In fact, there are very few data about nanoplastic concentrations in natural waters. We make the assumption, in a first estimation, that nanoplastic concentrations are the same order of magnitudes as small microplastics (1-1000 µm). Small microplastic concentrations determined by micro-spectroscopy are generally expressed in number of particles per volume and are typically between one to a few tens of particles per cubic meter (Koelmans et al., 2019) This corresponds, with a rough conversion, typically to mass concentrations between 0.1 and a few hundred of µg/L. The first concentration of nanoplastics in a river determined with pyrolysis-gas chromatography time of flight mass spectrometry were within this range (241.8 µg/L Sullivan et al., 2020)).

Koelmans, A.A., Nor, N.H.M., Hermsen, E., Kooi, M., Mintenig, S.M. et De France, J. (2019) Microplastics in freshwaters and drinking water: Critical review and assessment of data quality. Water Research 155, 410-422.doi: 10.1016/j.watres.2019.02.054

Sullivan, G.L., Gallardo, J.D., Jones, E.W., Hollliman, P.J., Watson, T.M. et Sarp, S. (2020) Detection of trace sub-micron (nano) plastics in water samples using pyrolysis-gas chromatography time of flight mass spectrometry (PY-GCToF). Chemosphere 249.doi: 10.1016/j.chemosphere.2020.126179

We have already argued the tested concentrations in the previous paper (Rowenczyk, L et al. Nanomat, 2021, 11(2): 482. DOI: 10.3390/nano11020482) where the effects were studied in algae. We think it unnecessary to justify it again here, it would weigh down the paper

We propose to add intro the introduction section « There is a considerable gap regarding the concentration of nanoplastics in environmental samples due to a lack of specificity and/or resolution of the current analysis methodologies. Only Sullivan et al. published a semi-quantitative data about polystyrene nanoplastics in the river Tawe (South Wales, 241.8 μg·L−1). »

  1. Since the difference in physical properties of nanoparticles affects the dispersibility, it is better to add the dispersive index in each exposure media. It is also known that the difference in shape affects the degree of toxicity.

The dispersibility index is not reliable for polydispersed populations like PSHA. It is very simple to discuss sizes for such a large audience.

Thus, the author discusses the possibility of difference in growth inhibition due to the fact that PS-HA is non-spherical, and the others are spherical, if possible.

Yes, but it is rather anecdotal. We propose to add in the discussion section « However, in the present study, size alone cannot explain here the observed differences in terms of effects. These results suggest that other parameters could influence their biological impact, such as the structure, shape and surface of the aggregates/particles. » 

  1. It is desirable to be able to compare and discuss growth inhibition for Chironomids and amphibians more quantitatively by calculating the EC50 of each microplastic.

The calculation of the EC50 for each nanoplastic does not seem to us to be a priority because there is no relation between the level of toxicity and the mechanism of toxicity which is discussed in the paper since it is the objective.

Reviewer 2 Report

See attachment

Author Response

This paper addresses a significant environmental challenge of our time and that is the issue of microplastics. Even though the impact of micro-and nano-particulate plastic on marine organisms has been addressed by a number of marine biologists, little has been done regarding the impact of the same on fresh water organisms. The authors address ecotoxicity of hetero-aggregates of polystyrene nanospheres in Chironomidae and Amphibians. The manuscript is straightforward with the main conclusion being that whereas some of these particles, particularly the PSHA, reduced growth rates, impacted developmental stages and genotoxicity, exposure of the organisms did not result in mortality during the experiments. My only concern was that the manuscript was too wordy and should be shortened to attract more readers.

Parts have been removed from the introduction for simplification et attracting more readers, as well as from the discussion section where some paragraphs have been modified for clarity. The english has been revised by an english speaker.

Minor points.
1) Scale bars should be placed on all the TEM micrographs in Table 1.

Scale bars have been placed on all the TEM micrographs

2) If possible the authors should show pictures of Chironomidae and Amphibian stages for those who may not be familiar with these organisms.

it seems important to us not to draw attention to animal experimentation, especially in higher vertebrates such as amphibians. this may displease some authors. we prefer not to put a photograph

Reviewer 3 Report

The discussed article is an interesting work, supplementing the available knowledge on the toxicity of nanoplastics.

In my opinion, the article can be published in its current form.

Round 2

Reviewer 1 Report

There is no comment.